# Mining the Biosynthetic Potential for Specialized Metabolism of a *Streptomyces* Soil Community

**DOI:** 10.3390/antibiotics9050271

**Published:** 2020-05-23

**Authors:** Matthieu Nicault, Abdoul-Razak Tidjani, Anthony Gauthier, Stéphane Dumarcay, Eric Gelhaye, Cyril Bontemps, Pierre Leblond

**Affiliations:** 1Université de Lorraine, INRAE, DynAMic, F-54000 Nancy, France; matthieu.nicault@univ-lorraine.fr (M.N.); abdoul-razak.tidjani@univ-lorraine.fr (A-R.T.); anthony.gauthier@inra.fr (A.G.); cyril.bontemps@univ-lorraine.fr (C.B.); 2Université de Lorraine, INRAE, IAM, F-54000 Nancy, France; eric.gelhaye@univ-lorraine.fr; 3Université de Lorraine, INRAE, LERMAB, F-54000 Nancy, France; stephane.dumarcay@univ-lorraine.fr

**Keywords:** *Streptomyces*, soil, specialized metabolism, biosynthetic gene cluster, community, diversity

## Abstract

The diversity and distribution of specialized metabolite gene clusters within a community of bacteria living in the same soil habitat are poorly documented. Here we analyzed the genomes of 8 *Streptomyces* isolated at micro-scale from a forest soil that belong to the same species or to different species. The results reveal high levels of diversity, with a total of 261 biosynthesis gene clusters (BGCs) encoding metabolites such as terpenes, polyketides (PKs), non-ribosomal peptides (NRPs) and ribosomally synthesized and post-translationally modified peptides (RiPPs) with potential bioactivities. A significant part of these BGCs (n = 53) were unique to only one strain when only 5 were common to all strains. The metabolites belong to very diverse chemical families and revealed that a large diversity of metabolites can potentially be produced in the community. Although that analysis of the global metabolome using GC-MS revealed that most of the metabolites were shared between the strains, they exhibited a specific metabolic pattern. We also observed that the presence of these accessory pathways might result from frequent loss and gain of genes (horizontal transfer), showing that the potential of metabolite production is a dynamic phenomenon in the community. Sampling *Streptomyces* at the community level constitutes a good frame to discover new biosynthetic pathways and it appears as a promising reservoir for the discovery of new bioactive compounds.

## 1. Introduction

Antimicrobial resistance is a major and global threat, with an estimated 33,000 human deaths per year in the European Union and 700,000 worldwide [1]. It is projected to be the leading cause of death by 2050 (with an estimated 10 million deaths globally). Finding new molecules is one of the solutions making it possible to challenge this scenario. The Actinobacteria are the most prolific providers of specialized metabolites, including two-thirds of all known antibiotics as well as many anticancer, antifungal, and immunosuppressive agents. These bacteria are thus of great importance in medicine, plant sciences, and biotechnologies [2]. Among Actinobacteria, the only genus *Streptomyces* provides by itself half of all known active compounds [3,4] and therefore constitutes a natural reservoir of potential new products to thwart antimicrobial resistant strain emergence. *Streptomyces* are soil dwelling bacteria also inhabiting marine and freshwater ecosystems. They generally have large genomes (6–12 Mb) organized in a linear chromosome [5]. They play an important role in the organic material recycling and are involved in symbiosis with plants, fungi, insects, and animals [6]. These interactions can be beneficial, producing compounds that protect the host against pathogens [7], or enzymes that degrade resistant polymers such as lignocellulose [8].

Though this potential of metabolite production has been known for a long time, recent genomic approaches have revealed that it was in fact largely underestimated, as a single *Streptomyces* strain can possess in average between 30 to 50 biosynthetic gene clusters (BGCs) that could encounter for 8–10% of its genome [2,9]. Most of them are cryptic in standard laboratory culture conditions but can be awakened by diverse approaches [10]. For instance, *Streptomyces ambofaciens* isolated from a French soil in the 1950s was historically known to only produce the drug spiramycin, used in human therapy as an antibacterial agent and for the treatment of toxoplasmosis; and the pyrrole-amide congocidine [11]. Mining of its genome allowed the identification of 23 clusters potentially involved in the production of other secondary metabolites [9,12] and different approaches including genome modification led to the identification of different new compounds, or already known compounds such as kinamycin angucyclinone antibiotics [13,14] or the stambomycins [15]. If various approaches exist to unravel the potential of metabolite production of the *Streptomyces* such as genome mining [16] and editing (e.g., ribosome engineering) [17], regulator manipulations [18], synthetic biology [19], variations in culture conditions (e.g., co-culture, OSMAC, and nutrient conditions screening) [20], most of these approaches rely on natural isolates. Thus, prospecting the *Streptomyces* diversity remains essential. Studies sampled *Streptomyces,* for instance, from underexplored environments [21] such as mangroves [22], in deserts [23], caves [24], in the oceans, or in association with plants [25], insects [26], or marine animals like sponges [27] to increase the probability to discover new compounds. Interestingly, new insights in *Streptomyces* diversity showed that strains barely distinguishable at the taxonomic level can have distinct specialized metabolisms [5,28,29,30,31]. Thus, it seems that a large diversity of metabolites can be produced at the intra-specific level. We recently showed that this diversity resulted from the plasticity of the *Streptomyces* genome which experienced massive gene fluxes and recombination events even over short evolutionary times [5]. 

Regarding this large diversity, we challenged in the present study the idea that the potential of biosynthetic pathways found in a small number of strains isolated from a standard temperate soil should reveal a potential of new molecules. We therefore isolated at a small spatial scale (cm) a *Streptomyces* community from grains of soil originating from a temperate French forest. We used genome-guided, metabolomic, and phenotypic analyses and revealed their potential to produce metabolites of interest. We therefore show that even in a grain of soil, the potential to discover a large diversity of secondary metabolites and among them new activities is a reasonable approach regarding the discovery of new drugs.

## 2. Results and Discussion

### 2.1. Isolation and Taxonomic Characterization of a Streptomyces Community

As already described, in Tidjani et al. [5], soil was sampled in the forest of Montiers (France). Eight grains of soil maximally distant of 8 cm and in the order of mm^3^ were taken for strain isolation. Forty-six sporulating isolates with an actinomycetal phenotype (filamentous growth and sporulated colonies) were selected and phylogenetically characterized with a five-gene multi locus sequence analysis (MLSA) scheme (Appendix A). All the isolates belonged to the *Streptomyces* genus and eight strains were retained for further analyses. The three strains (RLB1-8, RLB1-9, and S1D4-23) were already described and their genome sequenced in Tidjani et al. [32]. These strains were highly related and were part of the same population. Here, we added RLA2-12 as another representative strain of this population. The four additional strains (RPA4-5, RPA4-2, RLB1-33, and S1D4-11) were chosen in order to consider more taxonomically distant representatives of the community. After genome sequencing (see below), the 16S rDNA sequences of our isolates were extracted and used to specify their taxonomic position by a phylogenetic reconstruction in comparison with close reference strains (Figure 1A). These latter were identified by Blast search on NCBI © with the 16S rDNA sequence of our strains. The strain RLB1-33 was closely related to *S. mirabilis*, while RPA4-2 and S1D4-11 grouped together as sister group, but at the exclusion of any reference strain. The strains RLB1-8, RLB1-9, S1D4-23, and RLA2-12 had identical 16S rDNA sequences and were closely related to *S. olivochromogenes*. These seven strains belonged to clade II of the *Streptomyces* genus according to McDonald and Currie [33] classification. The strain RPA4-5 was more distant from the seven others and potentially belonged to the *S. platensis* species. This last strain belonged to the clade called “other” in McDonald and Currie classification which underlined the phylogenetic dispersion of our sampling. These results were confirmed by average nucleotide identity-based (ANIb) comparisons (Figure 1B).

Bioassay experiments (n = 104) were performed between all the pairs of strains, as well as with four strains isolated from the same soil and belonging to other bacterial genera (*Bacillus*, *Rhodococcus,* and *Paenisporosarcina*) (Appendix A). It showed that many strains had different growth inhibition patterns, with in the extreme cases, the strains S1D4-11 and RPA4-5 that inhibited all or almost all strains, and on the other hand, the strains RLB1-33, RLA2-12, and RLB1-8 that inhibited no indicator strain in the tested conditions. Interestingly these inhibition patterns could differ between closely related strains suggesting that, the metabolic profile could show some inconsistencies with phylogenetic relationships.

### 2.2. Genome Sequencing and BGC Identification

Genomic studies were performed on the eight selected strains; five of them were fully sequenced during the present study (Table 1). A combination of Nanopore (Oxford Nanopore Technologies) and Illumina technologies has been used. One to two large contigs covering the whole genome of each strain were obtained and enabled to acquire each linear chromosome in one scaffold and to identify one extrachromosomal element in strain RLA2-12 (predicted circular). The total genome sizes of the newly sequenced genomes ranged from 9.85 to 12.27 Mb, positioning these strains among the largest *Streptomyces* genomes and even among the largest bacterial ones (Table 1). The annotation was performed using the RAST tool kit (RASTtk, [34]) available on the Rapid Annotation using Subsystem Technology (RAST) platform. In order to estimate the potential of the *Streptomyces* community for the production of specialized metabolites, the genomes were screened in silico with antiSMASH [35] for the presence of biosynthetic gene clusters (BGCs) putatively coding for the production of such compounds (Figure 2). A total of 261 BGCs were detected across all strains, each with 25 to 36 BGCs with an average of 32. Regarding the overall community, a pool of 45 distinct BGCs were identified, with 26 shared by at least two strains and 19 unique to one strain. It should be noticed that the strain RPA4-5 which is the taxonomically more distant possessed more BGCs than the others with a ratio 3.8 BGCs per megabase of DNA versus 2.8-3 for the others. 

From the identified BGCs, the antiSMASH analysis enabled to group the predicted metabolites into six different classes that all possess a large diversity of bioactive molecules. Four of these classes encompassed peptides that are differentiated according to their biosynthesis enzymology. The NRPs are synthetized by non-ribosomal peptide synthases (NRPSs), the PKs by polyketide synthases (PKSs), the PKs/NRPs by hybrid biosynthesis pathways and the RiPPs are ribosomally synthetized and post-translationally modified peptides. The fifth group is constituted by terpenes and the last group named “other” gathers all the unassigned BGCs. These different classes were present and relatively frequent in all the strains (Figure 2). Aside from unassigned BGCs (10 to 15 per strains), terpenes (4–6 per strain), RIPPs (3–7 per strain), and NRPS (2–8 per strain) were the most frequent. 

In order to have a better assignment of these 261 BGCs, we compared them to the MIBiG database [36]. Among them, 68 showed strong similarities (over 70%) to known BGCs and therefore could potentially be involved in the synthesis of similar or analogous compounds. The 193 remaining clusters could be on the other hand at the origin of the synthesis of more diverse or new compounds. With the aim of finding new compounds, it is interesting to differentiate at the community scale highly conserved BGCs and BGCs that are only present in one or a few strains. The first ones are indeed more likely to be essential to the *Streptomyces* metabolism in their environment when the second may encode biosynthetic pathways involved in the production of novel bioactive compounds providing ecological advantages [37]. For this purpose, we constructed a similarity network of the BGCs among different genomes using BigScape ([38], Figure 3) which uses a modified Pfam domain similarity metric and represented the shared BGCs as a heatmap (Figure 4A). By doing this, 93 gene cluster families (GCFs) were detected of which 53 were strain specific. The more related the strains are, the more they share common biosynthesis pathways (Figure 4A). Although the BigScape analysis allows BGCs to be grouped into families, each family represents a large diversity of molecules deriving from related backbones. Hence differences such as the presence or absence of genes or point mutations can result in the formation of metabolites with distinct final structures. The presence of common biosynthetic pathways appears to follow phylogenetic links. If we only consider the closest representatives of the community (i.e., at the exclusion of the taxonomically distant RPA4-5) 13 GCFs are common, when only five (ectoine, hopene, desferrioxamines, spore pigment, and bacteriocin) are unanimously shared when RPA4-5 is considered. These five common GCFs may provide essential metabolites for *Streptomyces* to adapt to its biotic and abiotic environment. For example, the production of a soluble low-molecular-weight compound such as ectoine which counteracts the deleterious effects of salinity and loss of water on cell physiology [39] is shared by all eight strains. The “spore pigment” conferring a potential increased resistance to UV radiation [40] is also widely found. Hopanoids were also found in all the strains; the hopene BGC was present in all genomes while the albaflavenone biosynthetic pathway was present in all but RPA4-5. Although not essential for growth, these compounds impact membrane fluidity and permeability and were hypothesized to alleviate stress in aerial mycelia by decreasing water permeability of the cell membrane [41,42,43]. All the studied *Streptomyces* produced siderophores which play an important role in extracellular solubilization of iron and are essential for the growth and development of the bacterium. Three BGCs in each strain were tagged as siderophore biosynthetic pathways but only one was ubiquitous, that is desferrioxamines B and/or E. Sarpeptins A/B, myxochelins A/B or delfibactins A/B (NRPS, [44]) were also predicted in several strains. The volatiles geosmin and 2-methylisoborneol BGCs were found in 6 strains but absent from strain RPA4-5. Geosmin is famous for giving soil its familiar “earthy” smell. These two terpenoids have recently been implicated in the attraction of a soil arthropod facilitating their dissemination in the environment [45]. The RiPPs (ribosomally synthesized and post-translationally modified peptides) lassopeptides siamycin I, MS 271, and albusnodin, and RiPP lanthipeptides SapB and SAL-2242, were also identified. The carotene isorenieratene and the PKS products lydicamycin, alkylresorcinol, and pepticinnamin E also belong to the identifiable BGCs.

Specific GCFs may provide new biosynthetic pathways and offer opportunities to discover new compounds promising in the fight against antibiotic resistance in pathogenic bacteria. Studying their diversity and distribution may also help to understand the role of antibiotic in the ecological context. Hence, specific BGC may specifically help in the challenging biotic competition in soil, and help its owner or its conspecific to cope against a competitor. The contribution of the different strains of the community to the specific GCFs is illustrated by the heat-map (Figure 4B). Consistently with the previous results, the strain RPA4-5 which is the most distantly related strain provided alone 28 out of 53 of the unique pathways. Among them, two compounds were predicted by antiSMASH with high confidence (100% similarity): SapB and albusnodin (RIPPs). Strains RPA4-2 (16 BGCs), S1D4-11 (14 BGCs), and RLB1-33 (9 BGCs) also contributed a lot to the community repertoire. Recent studies have already pointed out that the correlation between the presence of BGCs and the taxonomy of *Streptomyces* is difficult to assess. It is the case at the genus level [31], but also between closely related species or at the intra-specific level [46,47,48]. These results can be explained by a high potential to gene exchange enabling the shuffling of the BGC repertoire and by the adaptive pressure applied in a specific niche to retain or to lose BGCs. The strains compared in other studies are generally isolated from various environments, blurring the niche effect. Here we investigated such presence/absence of BGCs among strains sharing the same habitat at different taxonomical levels. Despite this attention brought to the sampling scheme, it does not appear that genes are specifically conserved between distant strains and close phylogeny remains the main factor to share BGCs.

The fact that the number of specific GCFs increases with genetic distance suggests that gain and loss of BGCs are events occurring and cumulating along the evolutionary time leading to divergence. BGC replacement may occur even if the pressure is high on the function provided, especially when the newly acquired BGC performs the same function [49]. For example, as long as siderophores are produced and iron assimilation is ensured, the BGCs responsible for their synthesis can be replaced by horizontal transfer. On the other hand, other examples have shown that a few discrete differences may lead to phenotype differentiation between closely related strains. The polymorphism in *Bacillus subtilis* living in the same cubic centimeter impacted quorum sensing and led to kin differentiation [50]. Similarly, comparison of two closely related groups of *Myxococcus* strains enabled the identification of a 150-kb region involved in their merging phenotype [51]. Regarding specific BGCs, the turnover could even be much higher leading to a high diversification of the metabolites that can be used in biotic interactions such as antibiosis. As we have shown in a previous study [5], specific capacities to produce bioactive compounds such as antibiotics could be considered as public goods for the population. More distant strains forming a community could also share common goods like closely related strains belonging to the same population.

When the BGCs distribution is observed in the genetically closest strains, i.e., within the group of strains RLB1-8, RLB1-9, S1D4-23, and RLA2-12, it is possible to infer the events that led to the contemporary genomes. Thus, without being able to infer the genomic organization of the common ancestor of this group of strains, it is possible on the basis of parsimonious reasoning to retrace the events of gain and loss of BGCs occurring within this group. The BGCs present in a single strain were probably gained by horizontal transfer. Those absent from a single strain were probably lost. For all other cases, it is difficult to decide given the small number of strains considered. Only 26 of the 45 BGCs are shared by all strains despite their close phylogenetic proximity. Thirteen are shared by two or three strains, six are present in only one strain. This great diversity is concentrated in the terminal regions of the linear chromosome with the six specific BGCs present in the terminal 25% of the chromosome (not shown). In addition, with the exception of a single loss event located in the central region of the chromosome (loss of the lassopeptide BGC, siamycin BGC in strain RLA1-12), all loss or gain events occurred in the terminal regions. The same picture appears when BigScape GCF groupings are used to map BGCs that are common (or almost common) or specific to all the sample strains (Appendix A). Hence, while common BGCs are scattered more or less in the same order in the center of the chromosome, specific ones are mostly localized in the chromosome arms. All together, these results highlight the high potential of metabolite production encountered in this *Streptomyces* community that is supported by strain specific clusters.

### 2.3. Metabolite Profiling

In order to pursue the diversity characterization of the studied *Streptomyces* strains, a metabolomic approach was used. After growth on GA Petri dish and solvent extraction, the extracts were separated by GC/MS and analyzed through the GNPS workflow. GC-MS was already used for the characterization of bioactive *Streptomyces* strains [52]. The used derivatization allowed to detect more compounds like amino acids and lipids. For the eight strains taken together, it revealed the presence of 5945 features divided into 1310 unique categories (a feature corresponds to *m/z* with retention time as described in Tourneroche et al. [53]). A large majority of features (97%) was found in at least two strains and only 36 (3%) were found in only one strain (Appendix A). It has been shown that in laboratory culture conditions (i.e., pure culture in standard growth conditions), most of the accessory genes including BGCs, remained silent [10]. Thus, the fact that most of the metabolomic profile is shared between strains could reflect the experimental conditions and revealed mainly compounds of the primary metabolism. A specific metabolic pattern could be nevertheless observed for each strain (Figure 5). These results were consistent with a previous study performed by Antony-Babu et al. ([28]), who showed no obvious relationship between the metabolite profiles of 10 *Streptomyces* strains with identical 16S rDNA genes. The bioassays conducted with our strains revealed that some antimicrobial compounds were produced (Appendix A). These activities likely depend on the expression of a single BGC as we have shown recently in one of the studied strains (RLB1-9, [5]) where the antibacterial activity was correlated to the presence of a NRPS gene cluster not present in other strains. It is logical that these activities were not sufficient to differentiate the global metabolomes of our strains. We tried to identify the nature of the 36 specific features to one strain in using the GNPS databases (NIST and others), but no identification could be performed. Among the shared features, 87 were common to all the strains and eight of them could be identified as ethanolamine, cyclohexylamine, hexane, glycine, tocopherol, 7,(5-alpha)-cholesten-2-beta,3-beta-14-alpha-22R,25-pentol-6-one, 2-Aminoimidazole, and 2(1H)-Pyrazinone. These compounds are involved in lipid and amino-acid synthesis and could result from the primary metabolism. Interestingly, the two latter have been detected in some *Streptomyces* isolates as potent antibiofilm agent [54] or to have a cytotoxic activity against colorectal carcinoma cell lines [55]. 

## 3. Conclusions

Our study revealed a strong variability of the capacity to produce specialized metabolites among a sample of *Streptomyces* strains isolated from a forest soil micro-habitat. Only a few BGCs were found to be shared by all the isolates and likely ensure essential physiological traits. In contrast, a large part of the BGCs was strain-specific and constitutes a huge reservoir for new genes of interest. A preliminary metabolomic analysis revealed that although a specific profile can be assigned to each strain, most of the metabolome is shared, likely reflecting the expression of the primary metabolism. Further prospect will consider the global expression in different growth conditions (abiotic and biotic stresses including co-culturing) of the strains in order to reveal the expression of new metabolites correlate to the diversity of BGCs.

## 4. Material and Methods

### 4.1. Strains, Culture, and Storage

The *Streptomyces* were isolated from grains of soil on the order of cubic millimeters in size from a clod of soil collected in the Montiers-sur-Saulx forest in France (GPS coordinates: 48°32′37.248″N, 5°18′21.946″E) as described in Tidjani et al. [5]. Non-*Streptomyces* bacteria were isolated from the same forest soil during another sampling campaign. *Streptomyces* were grown on GA medium (15 g starch, 5 g NaCl, 5 g KCl, 0.5 g K2HPO4, 0.5 g KNO_3_, 0.5 g MgSO_4_, 15 mg FeSO_4_, and 15 g Bacto Agar into 1 L of water adjusted in pH 7.2) and all the other bacteria on LB medium (10 g tryptone, 5 g yeast extract, 10 g NaCl for 1 L) at 30 °C. *Streptomyces* were stored at −80 °C as spore suspensions in glycerol [56]. For metabolite extractions, the initial spore stock of *Streptomyces* was diluted to give a solution of 10^5^ spores/ml. Ten microliters of this solution were then spread over a Petri dish. Cultures (3 replicates) were grown for 14 days and cut out with a spatula and gathered into a flask for ethyl acetate extraction (1/1 v/v). This mixture was shaken during 1 h 30 min and filtered to recover the liquid. After extraction, the organic solvent was evaporated under vacuum using a rotary evaporator. Dried extracts were stored at −18 °C before analysis.

### 4.2. Phylogenetic Analyses

MLSA genes (*atpD*, *gyrB*, *recA*, *rpoB*, and *trpB*) were amplified using primers and PCR conditions described by Guo et al. [57], respectively. Evolutionary analyses were conducted in MEGA7 [58] (Kumar 2016). The MLSA tree (3178 concatenated positions) was built by Neighbor-Joining with a Kimura’s two-parameter distance correction. After removing positions containing gaps in the alignment, the 16S rDNA tree was built by Maximum Likelihood based on the General Time Reversible model with 1432 positions. Branch likeliness was calculated with bootstrap analyses (100 replicates). ANIb values were calculated with Jspecies (http://jspecies.ribohost.com/jspeciesws/) [59].

### 4.3. Bioassays

All inhibition growth experiments were performed on GA medium [60]. Inhibition growth experiments between *Streptomyces* were performed by streaking the tester *Streptomyces* from a single colony to form a line on a plate and were incubated at 30 °C for 5 days. The receiver strain was then applied perpendicularly to the tester and incubated at 30 °C. The tester strain was considered inhibitory when it prevented the growth of the receiver streak in its vicinity. Inhibitory growth experiment against *Bacillus*, *Paenisporosarcina* and *Rhodococcus* were performed in inoculating for 5 days a 5 µL drop (10^7^ spores per mL) of a *Streptomyces* spore suspension. After an overnight pre-culture, the receiver strains were overlaid in LB soft (0.4% agar) at a 600nm OD of 0.05 on top of the *Streptomyces*. Their growth inhibition was monitored by the observation of an inhibitory halo.

### 4.4. Genome Analysis

The *Streptomyces* strains used in this work have been completely sequenced and assembled. Their accession number corresponding to their genome sequences are indicated in Table 1. Sequencing and assembling were performed via the I2BC NGS platform (France). Automatic annotation of the genome sequences was achieved using the RAST tool kit (RASTtk, [34]) available on the Rapid Annotation using Subsystem Technology (RAST) platform with a CDS size threshold of 50 amino-acids. Further, the annotation files were used as input of antiSMASH [35] in order to predict the BGC content of each genome. A similarity network of the BGCs was obtained using BigScape which uses a modified Pfam domain similarity metric implemented [38,61]. A cut-off of 0.75 was used for the analysis.

### 4.5. GC/MS Metabolite Profiling

Dried extracts into glass vials were silylated by adding 50 μL of BSTFA/TMSCl (99/1) and heated at 50 °C for 12 h. Then, the derivatizing reagent was evaporated, the extract derivatives were diluted in ethyl acetate and transferred for gas chromatography analysis. GC-MS analysis was performed on a Clarus 680 gas chromatograph coupled to a Clarus SQ8 quadrupole mass spectrometer (Perkin Elmer Inc., USA). Gas chromatography was carried out on a 5% diphenyl /95% dimethyl polysiloxane fused-silica capillary column (DB-5ms, 30 m × 0.25 mm, 0.25-μm film thickness, J&W Scientific, USA) with helium as carrier gas at a constant flow of 1 mL/min. The gas chromatograph was equipped with an electronically controlled split/splitless injection port. The injection (1 μL) was performed at 250 °C in the splitless mode. The oven temperature program was as follows: 80 °C for 2 min period, increase from 80 °C to 190 °C at a rate of 10 °C/min, increase from 190 °C to 280 °C at a rate of 15 °C/min and hold for 5 min, then 10 °C/min until 300 °C and hold for 14 min. Ionization was achieved under the electron impact mode (70 eV ionization energy). The source and transfer line temperatures were 180 °C and 250 °C, respectively. Detection was carried out in scan mode: m/z = 45 to m/z = 700. The detector was switched off in the initial 2 min (solvent delay).

### 4.6. Statistical Analysis of Metabolite Profiles

Obtained raw data were converted to the appropriate format using Mzmine2 software [62] Converted data were analyzed using GNPS workflow for GC/MS data [63]. Some of the identified compound were analyzed by using KEGG platform in order to verify the involved metabolism [64]. With these data an absence/presence table of features was obtained. All statistical analyses and heat maps have been performed using the R software with package “pheatmap” [65].

## Figures and Tables

**Figure 1 antibiotics-09-00271-f001:**
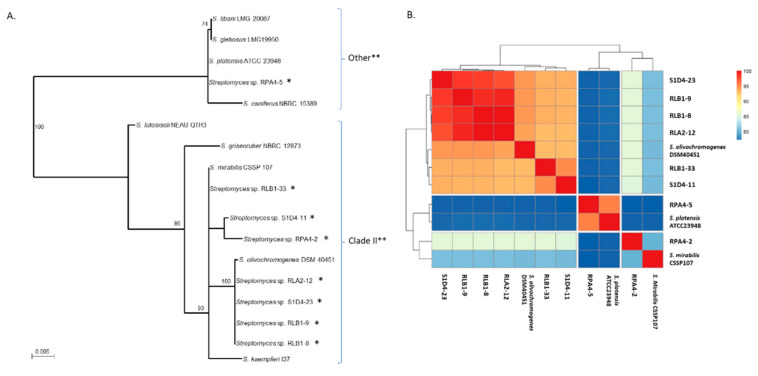
Genetic relationships between *Streptomyces* isolates of the community. (**A**). 16S rRNA phylogenetic tree of strains. The tree was based on a 1432 nucleotide alignment of the sequenced strains (marked with a * symbol) and closely related reference *Streptomyces*. The tree was built using the Maximum Likelihood method and with 100 bootstrap replicates. Bootstrap values below 70 were not represented. The scale represents mutations per nucleotide. The ** symbol represent the *Streptomyces* clades as defined by McDonalds and Currie ([33]). (**B**). Average nucleotide identity (ANI) between all pairs of genomes is represented as a heat map. According to taxonomic standards, strains that belong to a same species have identity scores higher than 95% (i.e., in red in the heat map).

**Figure 2 antibiotics-09-00271-f002:**
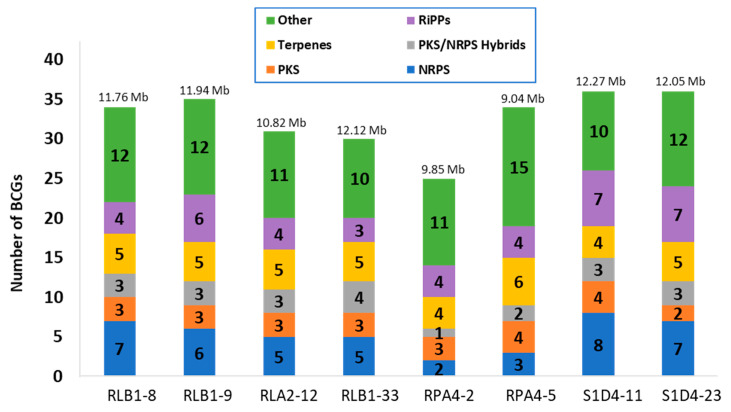
Distribution of biosynthetic gene clusters (BGCs) in strains. The identification of BGCs and their classification into families was performed by antiSMASH. Each family is represented by a color code and their numbers in each strain is indicated in the bar. The genome size of each strain is indicated on the top. PKS: polyketide, NRPS: nonribosomal peptide, RiPPs: ribosomally synthetized and post-translationally modified peptide.

**Figure 3 antibiotics-09-00271-f003:**
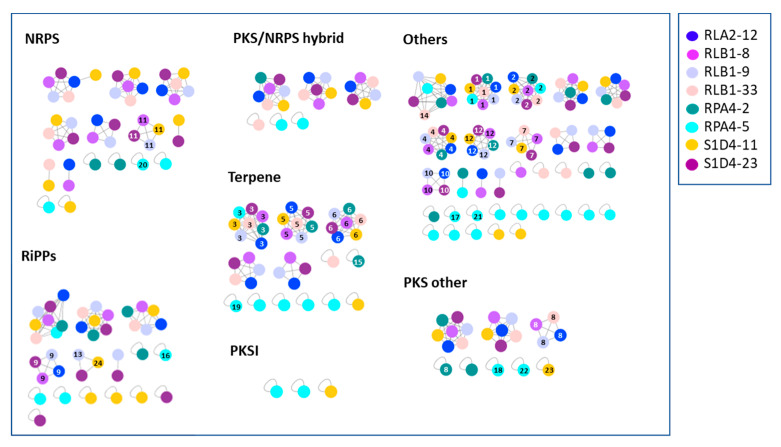
Similarity network of the predicted biosynthetic gene clusters (BGCs). After in silico prediction, BGCs shared by our 8 strains were represented in networks were similar BGCs between different strains are connected by a line. BGCs unique to one strain are symbolized by a singleton. Each strain is represented according to the color code. Numbered BGCs correspond to BGCs whose final product can be potentially predicted. 1, desferrioxamin B/E; 2, ectoin; 3, hopene; 4, spore pigment hybrid; 5, geosmin; 6, albaflavenone; 7, 2-methylisoborneol hybrid; 8, alkylresorcinol; 9, lanthipeptide A; 10, pepticinnamicin E; 11, diisonitrile antibiotic SF2768; 12, melanin; 13, siamycin I; 14, sarpeptine A/B; 15; 2-methylisoborneol; 16, albusnodin; 17, lydicamycin; 18, JBIR-76/77; 19, isorenieratene; 20, deimino-antipain; 21, lanthipeptide hybrids; 22, spore pigment; 23, lugdunomycin; 24, MS-271 (lassopeptide).

**Figure 4 antibiotics-09-00271-f004:**
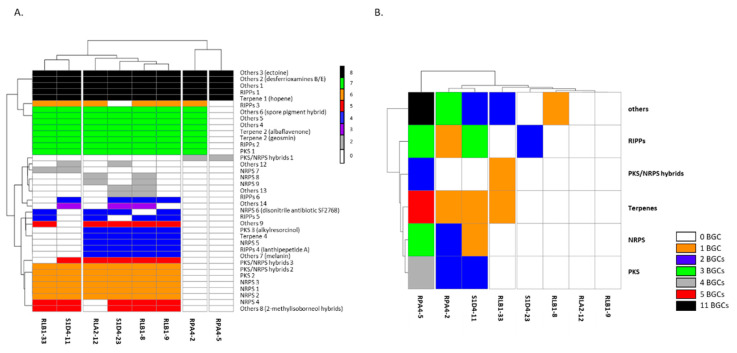
Distribution of gene cluster families (GCFs). (**A**). The presence of the different gene cluster families (n = 37) predicted by BigScape for the whole community is represented for each strain. BGCs are named according to the BigScape nomenclature. The absence of a GCF in a strain is represented by a white square and its presence by a colored one. The color of the square indicated the number of strains sharing this GCF. (**B**). Heat map of the number of unique GCFs brought by each individual strain of the community. The GCFs were grouped into 6 large families (NRPS, PKS, PKS/NRPS hybrids, RiPPs, Terpenes, others) and the color code indicates the number of unique GCFs present in a strain.

**Figure 5 antibiotics-09-00271-f005:**
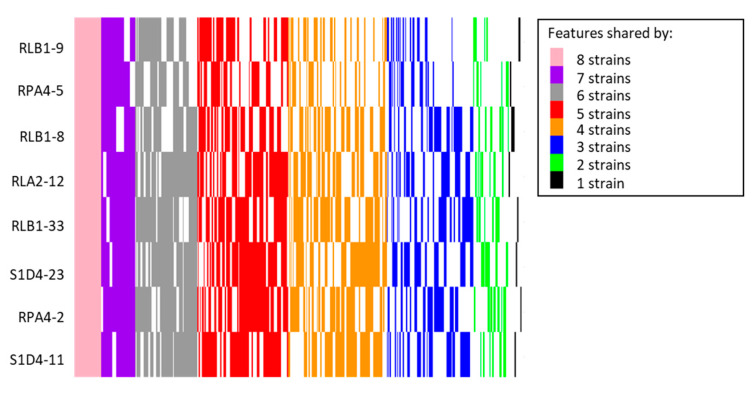
Metabolite profiling comparison. The presence/absence of all the features detected by GC/MS metabolite profiling (5,945 features grouped into 1,310 categories for the 8 *Streptomyces*) were compared in a heat map. A line represents each category and its color represents the number of strains that shares it.

**Table 1 antibiotics-09-00271-t001:** Genome characteristics and accession numbers of studied *Streptomyces* strains.

Strain/Plasmid	Replicon Size	Genome Size	CDS *	GC%	Genbank Accession Number
RLB1-8	11,765,395	11,765,395	10,891	70.2	*NZ_CP041650*
RLB1-9	11,940,408	12,201,201	11,085	70.2	*NZ_CP041654*
pRLB1-9.1 ^C^	154,158	-	181	69.0	*NZ_CP041653*
pRLB1-9.2 ^L^	106,635	-	120	68.7	*NZ_CP041652*
RLA2-12	10,825,588	10,892,946	10,031	70.3	JABAQG000000000
pRLA2-12.1 ^C^	67,358	-	85	69.8	-
S1D4-23	12,057,750	12,057,750	11,174	70.2	*NZ_CP041613*
RLB1-33	12,127,650	12,127,650	11,381	70.0	CP050974
RPA4-2	9,856,149	9,856,149	9287	70.9	CP050975
S1D4-11	12,276,515	12,276,515	12,065	69.9	CP051010
RPA4-5	9,047,156	9,047,156	9260	70.9	CP050976

* A threshold of 50 aa was applied after Rapid Annotation using Subsystem Technology (RAST) automatic annotation. ^C,L^: circular and linear. Accession numbers in italics correspond to previously reported genomes, straight numbers indicate newly sequenced genomes.

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
