# Peer review of "Mining the Biosynthetic Potential for Specialized Metabolism of a Streptomyces Soil Community"

_antibiotics, 2020, doi:10.3390/antibiotics9050271_

Round 1
Reviewer 1 Report
The manuscript entitled “Mining the biosynthetic potential for specialized metabolism of a Streptomyces soil community” by Nicault et al gives an overview on overview on the genomes of soil Streptomyces strains. They analyzed the genomes of 8 Streptomyces strains. Actinomycetes and in particular members of the genus Streptomyces are well known for their capacity to produce diverse natural products of biomedical and pharmacological relevance. Moreover, they evaluated the potential metabolic capacity of the strain to produce diverse classes of natural products using the current bioinformatics pipelines.
The study is very well written and well designed.
If possible, the strain should be deposited in a publicly accessible culture collection with the collection and strain number stated in the manuscript. The conclusions part is very poor, should be expanded

Author Response
The results and the discussion are mixed in such a way that the latter can appear poor.
We have reinforced the discussion points notably through the responses to expert comments 2 and 3.
We did not deposit the strains in culture collection yet, waiting for further investigation about the potential of these strains. However, the genomes and raw sequencing data are already available in the public databases giving the opportunity to explore their content. It is also always possible to have access to the strains by establishing a Material Transfer Agreement (MTA) between both research organisms.
Reviewer 2 Report
Mining the biosynthetic potential for specialized metabolism of a Streptomyces soil community by Nicault et al. is an outstanding look at the question of genetic diversity amongst Streptomyces isolated on a small scale from the same soil samples. It is a most welcome addition to a field that has not paid enough attention to this important question, and that has sometimes (often) been more concerned with discovery of new metabolites at the expense of careful study of the chemical ecology and function of these metabolites in the environment. The well-presented analysis in this paper, which addresses the commonalities and diversity in the genomes and metabolomes from strains isolated from the same soil is an important development for the field as a whole; the discussion of horizontal gene transfer of these biosynthetic gene clusters is particularly important. I hope that it will inspire other authors to also look at the genetic diversity and ecology of Streptomyces spp. isolated from the environment at the micro scale.
There are a few points that I think would benefit from a slightly more thorough discussion:
Major points
1. Are there any quantitative data available for the bioassays? (Table S2) Although not necessary for the conclusions of this paper, it would be interesting to discuss whether e.g. more closely related strains are more/less tolerant to the anti-microbial activity(activities) produced by their neighbours. The binary yes/no data presented in Table S2 may overlook some biological complexity.
2. The discussion of the conserved metabolites (Figs 3 and 4 and lines 140-161) is absolutely fascinating. I wonder if the authors could comment on how closely the observed phenotypes for these strains align with the bioinformatic comparisons? (e.g. for BGC4 in Figure 3, do all strains produce the same color spore pigment or do they have slightly different alleles, leading to different pigment colours? Further analyses of the other conserved metabolites would also be fascinating but is obviously outwith the scope of this paper; however, spore pigment colour can be readily visually observed.)
Detailed discussion of each biosynthetic gene cluster is obviously beyond the scope of this paper; however, it is important to note that addition or subtraction of a single gene to/from a BGC (or even key mutation(s) in a single gene) could lead to production of a different compound (with potentially different properties). Some discussion accompanying Figs 3 and 4, of whether the BGCs conserved in different strains lead to the production of the exact same molecule(s), or not, would seem to be appropriate.
3. The authors point out that resistant non-producing organisms may be able to take advantage of metabolite production by other members in a community (lines 188-190), a concept which deserves a little more attention. Are the authors suggesting that some strains may both carry the same BGC (with one expressing the genes in the cluster & producing the compound, and the other expressing only the genes encoding resistance? Or that there are genes conferring resistance located elsewhere on the genome (outwith the BGC)? Or…?
Obviously, a detailed analysis of the antibiotic resistance genes present in the genomes of these sympatric isolates is outside the scope of this paper; however, a slightly expanded discussion here would be most welcome. This could be more explicitly linked to some of the data presented in Table S2 (although some care must be taken as in many cases the BGCs or resistance genes might be present but not be expressed under the conditions tested).
4. Could the authors perhaps give a more expansive discussion synthesizing the results presented in Figures 4 and 5, and S2? There is some discussion of the antimicrobial compound produced by RLB1-9 (lines 220-3) but a number of other isolates show antimicrobial activities in Figure S2, which may be worth exploring in a little more detail.
5. In the discussion, the authors should broaden the scope of their conclusions by comparing their results to those from studies that have looked at genetic and phenotypic diversity in other bacterial species sampled from soil on mm- and cm- scales. (This has certainly been done for Myxococcus xanthus - perhaps for other soil-dwelling bacteria, as well, though I am less familiar with the literature for e.g. Bacillus or other species.)
Minor points
1. To compare the average nucleotide identities amongst these strains, it may be better to use ANIm instead of ANIb (as ANIb fragments the genome before aligning, so non-homologous regions may contribute by proxy through neighbouring sequence).
2. Figure 2 and the anti-SMASH analysis of the genomes: For the sake of clarity, could the authors please specify whether the anti-SMASH analysis was performed on the entire genome of each strain (including plasmids), or whether the three plasmids were excluded from this analysis? (From the genome sizes in Table 1, it appears to be the latter?)
3. Lines 220-1: “The bioassays conducted with our strains revealed that some compounds were produced in laboratory conditions” should perhaps read “…that some antimicrobial compounds…?” (as clearly all of the compounds observed were produced in laboratory conditions)
4. Materials and methods, line 244-5: “Three separated bacterial cultures were performed during 14 days” is not clear as written – what is meant by separated? (3 replicates?) how long was each culture grown? (all in parallel for 14 days?)
5. Requires editing throughout for minor grammatical mistakes – I have highlighted a few below but this is not an exhaustive list:
a. When using pronouns, make sure it is clear what they refer to – for example:
1. Abstract, line 20: “they exhibited a specific metabolic pattern” is confusing (grammatically, “they” would seem to refer to the metabolites in the previous clause – or perhaps it refers to the strains?)
2. Line 60: “We recently showed that it resulted” – what is “it”? same sentence, line 62 “their BGC contents” – whose?
3. Line 103: “that work” – which work?
b. Line 27 - be clearer by what is meant by “large genomes”
c. Line 96: “all or almost strains” à “all or almost all strains”
d. Lines 98-99: sentence unclear as written (“regarding their taxonomic relationship”?)
Author Response
Mining the biosynthetic potential for specialized metabolism of a Streptomyces soil community by Nicault et al. is an outstanding look at the question of genetic diversity amongst Streptomyces isolated on a small scale from the same soil samples. It is a most welcome addition to a field that has not paid enough attention to this important question, and that has sometimes (often) been more concerned with discovery of new metabolites at the expense of careful study of the chemical ecology and function of these metabolites in the environment. The well-presented analysis in this paper, which addresses the commonalities and diversity in the genomes and metabolomes from strains isolated from the same soil is an important development for the field as a whole; the discussion of horizontal gene transfer of these biosynthetic gene clusters is particularly important. I hope that it will inspire other authors to also look at the genetic diversity and ecology of Streptomyces spp. isolated from the environment at the micro scale.
There are a few points that I think would benefit from a slightly more thorough discussion:
Major points
- Are there any quantitative data available for the bioassays? (Table S2) Although not necessary for the conclusions of this paper, it would be interesting to discuss whether e.g. more closely related strains are more/less tolerant to the anti-microbial activity(activities) produced by their neighbours. The binary yes/no data presented in Table S2 may overlook some biological complexity.
We do not have quantitative data for the bioassays, as it is generally not relevant with the streaking method used between Streptomyces. We agree that it might overlook some biological complexity, but these data were mainly part of strain screening.
On the other hand, our binary results seem to show that strains are less inhibitory at the intraspecific level (e.g. strains RLB1-9, RLB1-8, S1D4-23 and RLA2-12), but we think that our data are too scarce at this stage to conclude, and a more representative sampling would necessary to correctly answer the question. For these reasons, we do not prefer to comment such results in this paper.
- The discussion of the conserved metabolites (Figs 3 and 4 and lines 140-161) is absolutely fascinating. I wonder if the authors could comment on how closely the observed phenotypes for these strains align with the bioinformatic comparisons? (e.g. for BGC4 in Figure 3, do all strains produce the same color spore pigment or do they have slightly different alleles, leading to different pigment colours? Further analyses of the other conserved metabolites would also be fascinating but is obviously out with the scope of this paper; however, spore pigment colour can be readily visually observed.)
We agree that having correlation between phenotypes and genome analysis would be interesting, but this information is not necessary for our message. Regarding the spore pigment coloration, no difference was observed between the different strains. Nevertheless, establishing correlation between colonial phenotype and the production of metabolites is difficult. The pigmentation of hyphae and spores results from the synthesis and accumulation of numerous pigmented compounds, and the analysis of only one BGC related to spore pigment won’t explain the observed phenotype if any.
Detailed discussion of each biosynthetic gene cluster is obviously beyond the scope of this paper; however, it is important to note that addition or subtraction of a single gene to/from a BGC (or even key mutation(s) in a single gene) could lead to production of a different compound (with potentially different properties). Some discussion accompanying Figs 3 and 4, of whether the BGCs conserved in different strains lead to the production of the exact same molecule(s), or not, would seem to be appropriate.
A sentence was added line 146-149.
‘Although the BigScape analysis allows BGCs to be grouped into families, each family represents a large diversity of molecules deriving from related backbones. Hence differences such as the presence or absence of genes or point mutations can result in the formation of metabolites with distinct final structures.’
- The authors point out that resistant non-producing organisms may be able to take advantage of metabolite production by other members in a community (lines 188-190), a concept which deserves a little more attention. Are the authors suggesting that some strains may both carry the same BGC (with one expressing the genes in the cluster & producing the compound, and the other expressing only the genes encoding resistance? Or that there are genes conferring resistance located elsewhere on the genome (outwith the BGC)? Or…?
Obviously, a detailed analysis of the antibiotic resistance genes present in the genomes of these sympatric isolates is outside the scope of this paper; however, a slightly expanded discussion here would be most welcome. This could be more explicitly linked to some of the data presented in Table S2 (although some care must be taken as in many cases the BGCs or resistance genes might be present but not be expressed under the conditions tested).
‘As members of the community coexist in the same habitat, a similar parallel could be drawn where resistant, but not producing strains can take advantage of metabolite production within the community.’
This sentence was ambiguous and probably led to mislead the referee. Our idea was to make a parallel between the functioning of a population and a community through the sharing of public goods. Regarding referee’s remarks, we cannot interpret and easily identify the resistance genes in BGCs and strains.
We clarified this point by rephrasing line 205-206.
- Could the authors perhaps give a more expansive discussion synthesizing the results presented in Figures 4 and 5, and S2? There is some discussion of the antimicrobial compound produced by RLB1-9 (lines 220-3) but a number of other isolates show antimicrobial activities in Figure S2, which may be worth exploring in a little more detail.
Establishing a correlation between an antibacterial activity and the presence of gene cluster is difficult to assess. Indeed, as mentioned before (remark 2), GCFs encompass a diversity of BGCs (point mutations, presence/absence of genes) that could impact the bioactivity. Further, differences in regulation can induce a differential expression. Moreover, most of them remain cryptic in standard growth conditions.
Establishing a clear correlation requires the construction of defective mutants, as we did, regarding RLB1-9, in a previous work. Such experiments are long and, in our case, hazardous as we do have many candidate BGCs to mutate and is out of the aim of this article.
- In the discussion, the authors should broaden the scope of their conclusions by comparing their results to those from studies that have looked at genetic and phenotypic diversity in other bacterial species sampled from soil on mm- and cm- scales. (This has certainly been done for Myxococcus xanthus - perhaps for other soil-dwelling bacteria, as well, though I am less familiar with the literature for e.g. Bacillus or other species.)
We added some comments lines 197-201.
Minor points
- To compare the average nucleotide identities amongst these strains, it may be better to use ANIm instead of ANIb (as ANIb fragments the genome before aligning, so non-homologous regions may contribute by proxy through neighbouring sequence).
ANIb and ANIm are often considered as good, and ANIb is often preferred as ANIm is less resolutive for distant species (<75% ANIb) (Liu et al. 2014 Antonie van Leeuwenhoek 107(1) DOI: 10.1007/s10482-014-0322-1.) It is not the case here but still we do think that having ANIm values would not improve or change our conclusions.
- Figure 2 and the anti-SMASH analysis of the genomes: For the sake of clarity, could the authors please specify whether the anti-SMASH analysis was performed on the entire genome of each strain (including plasmids), or whether the three plasmids were excluded from this analysis? (From the genome sizes in Table 1, it appears to be the latter?)
For these analysis, direct annotation file from RAST analysis were used. Furthermore, no changes were observed in our results when we started from DNA fasta file or from genbank annotated files.
antiSMASH analysis was also conducted on plasmid sequences and did not lead to any BGC identification.
- Lines 220-1: “The bioassays conducted with our strains revealed that some compounds were produced in laboratory conditions” should perhaps read “…that some antimicrobial compounds…?” (as clearly all of the compounds observed were produced in laboratory conditions)
Corrected.
- Materials and methods, line 244-5: “Three separated bacterial cultures were performed during 14 days” is not clear as written – what is meant by separated? (3 replicates?) how long was each culture grown? (all in parallel for 14 days?).
Rephrased.
- Requires editing throughout for minor grammatical mistakes – I have highlighted a few below but this is not an exhaustive list:
- When using pronouns, make sure it is clear what they refer to – for example:
- Abstract, line 20: “they exhibited a specific metabolic pattern” is confusing (grammatically, “they” would seem to refer to the metabolites in the previous clause – or perhaps it refers to the strains?)
Clarified
- Line 60: “We recently showed that it resulted” – what is “it”? same sentence, line 62 “their BGC contents” – whose?
Rephrased
- Line 103: “that work” – which work?
Clarified
- Line 27 - be clearer by what is meant by “large genomes”
Specified
- Line 96: “all or almost strains” à “all or almost all strains”
Done
- Lines 98-99: sentence unclear as written (“regarding their taxonomic relationship”?)
Rephrased
Reviewer 3 Report
In the manuscript, Nicault and colleagues present an interesting idea that new molecules can be potentially discovered by analysing a number of strains isolated from a standard temperate soil in previous studies. Using their genome sequence, a multi locus-based phylogenetic analysis and an antiSMASH-based analysis of biosynthetic gene clusters were conducted. This approach has appeared to be a popular strategy these days to show biosynthetic potential of soil bacteria – but afraid to say that it likely does not offer anything new. Moreover, I did not see the potential of new metabolites being presented in the manuscript, in part due to the major work presented here is apparently bioinformatic analysis. I also did not see the main figures included in the manuscript (the figure captions present at least in a copy I downloaded). At this stage, I can still see the potential of this manuscript being published in the journal, for example if the authors include additional data to increase the significance and if the authors describe furthermore into their GC-MS analysis. Detailed suggestions to improve this manuscript are listed below.
Broad comments:
- If GC-MS is strategically planned for this study, (i) please include the rationale behind using GC-MS as opposed to LC-MS, which perhaps is relatively more often used in microbial metabolite analysis. (ii) Is there any connection can be made between metabolomic data and bioassay data? While I realise highlighting the GC-MS application maybe does not reflect to their original idea, I thought it might increase a bit of the potential impact.
- Based on the caption of Figure 3, there are BGCs unique to one strain. (i) How unique are these clusters in comparison with known clusters? It seems strain RPA4-5 has some novel biosynthetic potentials, stemming from its distinct phylogenetic traits and biological activities. (ii) Line 333: Are these final products predicted with 100% values in antiSMASH?
- It is interesting that a great BGC diversity is discovered in the terminal of chromosomes (Lines 200-201). Are these also shared or unique among the 8 isolates at some levels? I would strongly suggest showing the data (in the supplementary data?)
- Bioinformatic data seemed to suggest a strong variation of biosynthetic capacity among the strains, whereas metabolomic data presented 3% variation of features detected. Were the GC-MS features excluding possible different MS adducts? Further discussion is warranted to address this point.
- While it is effective to summarize bioassay data into a table (e.g. Table S2), displaying a representative image for growth inhibition observed during experiment would be beneficial.
Specific comments:
In general, another careful read-through is encouraged for this manuscript.
Abstract (lines 16): 53 is more than half of what?
Line 74: The sentence is not clear
Line 75: What does actinomycetal phenotype look like?
Line 76: Introduce what MLST constitutes here as presented in the first time in the manuscript. Similar with GA in line 209.
Lines 177-179 and 204-205: ambiguous sentences
Lines 290: add “period” after “2 min”
Where is Table S1?
Typos can be found in the manuscript, e.g. GFC
Author Response
Ref 3.
In the manuscript, Nicault and colleagues present an interesting idea that new molecules can be potentially discovered by analysing a number of strains isolated from a standard temperate soil in previous studies. Using their genome sequence, a multi locus-based phylogenetic analysis and an antiSMASH-based analysis of biosynthetic gene clusters were conducted. This approach has appeared to be a popular strategy these days to show biosynthetic potential of soil bacteria – but afraid to say that it likely does not offer anything new. Moreover, I did not see the potential of new metabolites being presented in the manuscript, in part due to the major work presented here is apparently bioinformatic analysis. I also did not see the main figures included in the manuscript (the figure captions present at least in a copy I downloaded). At this stage, I can still see the potential of this manuscript being published in the journal, for example if the authors include additional data to increase the significance and if the authors describe furthermore into their GC-MS analysis. Detailed suggestions to improve this manuscript are listed below.
We do not understand why the referee could not upload and access to the figures which were provided as separated files (as the supplementary data) as recommended at the submission step.
Broad comments:
- If GC-MS is strategically planned for this study, (i) please include the rationale behind using GC-MS as opposed to LC-MS, which perhaps is relatively more often used in microbial metabolite analysis. (ii) Is there any connection can be made between metabolomic data and bioassay data? While I realise highlighting the GC-MS application maybe does not reflect to their original idea, I thought it might increase a bit of the potential impact.
A sentence was added line 229-231.
- The direct connection between bioassay, genomic and metabolomic data can obviously not be done at this stage and is only a long-term goal for a bio-guided approach, which is out of the scope of this article.
- Based on the caption of Figure 3, there are BGCs unique to one strain. (i) How unique are these clusters in comparison with known clusters? It seems strain RPA4-5 has some novel biosynthetic potentials, stemming from its distinct phylogenetic traits and biological activities. (ii) Line 333: Are these final products predicted with 100% values in antiSMASH?
- In contrast in RPA4-5, 4 BGCs are 100% similar to known clusters with product predicted with a high confidence: ectoine, desferrioxamine which are shared by all the strains and RIPPs albusnodin and SapB which are strain-specific.
This information was added lines 180-181.
- legend figure 2: the antiSMASH similarity values do not apply in this figure since only the main types of pathway are described here.
- It is interesting that a great BGC diversity is discovered in the terminal of chromosomes (Lines 200-201). Are these also shared or unique among the 8 isolates at some levels? I would strongly suggest showing the data (in the supplementary data?)
A figure showing the common and specific BGCs was added in the supplementary data (Figure S3).
Lines 219-222 ‘Hence, while common BGCs are scattered more or less in the same order in the center of the chromosome, specific ones are mostly localized in the chromosome arms.’
- Bioinformatic data seemed to suggest a strong variation of biosynthetic capacity among the strains, whereas metabolomic data presented 3% variation of features detected. Were the GC-MS features excluding possible different MS adducts? Further discussion is warranted to address this point.
This point has been discussed from line 233: “A large majority of features (97%) was found in at least two strains and only 36 (3%) were found in only one strain (Fig. S2). It has been shown that in laboratory culture conditions (i.e. pure culture in standard growth conditions), most of the accessory genes including BGCs, remained silent [50]. Thus, the fact that most of the metabolomic profile is shared between strains could reflect the experimental conditions and revealed mainly compounds of the primary metabolism.”
Here, we worked with an electronic impact ionization mode. With this mode, adducts (acids, salts) are not detectable.
- While it is effective to summarize bioassay data into a table (e.g. Table S2), displaying a representative image for growth inhibition observed during experiment would be beneficial.
An additional figure was added in the supplementary data (Figure S4).
Specific comments:
In general, another careful read-through is encouraged for this manuscript.
Abstract (lines 16): 53 is more than half of what?
‘Of these BGCs’ added
Line 74: The sentence is not clear
Rephrased
Line 75: What does actinomycetal phenotype look like?
Rephrased
Line 76: Introduce what MLST constitutes here as presented in the first time in the manuscript. Similar with GA in line 209.
MLST was replaced by MSLA and the acronym developed.
GA is a medium reported in Abrudan et al. but we can’t find the correspondence of the acronym.
Lines 177-179 and 204-205: ambiguous sentences
Both sentences were clarified.
Lines 290: add “period” after “2 min”
Done
Where is Table S1?
corrected
Typos can be found in the manuscript, e.g. GFC
Done
Round 2
Reviewer 1 Report
much improved
Author Response
There is no remark to be answered.
Reviewer 3 Report
Thank you for the making the effort to revise the manuscript. The main figures have been provided and included now. The figures help me to better understand the story. However, those figures need some work to improve resolutions and readability (e.g. some figure texts are not readily readable). Related to the figures, there are additional comments or questions:
- Figure 1. (i) How comparable are phylogenetic trees generated based on 16S rRNA gene sequence versus multilocus housekeeping genes as applied in the reference 33? (ii) Is there any reason not to include an outgroup?
- Figure 4. Description on how to determine GCFs as unique ones would be beneficial to have in the manuscript.
Specific comments:
In general, data visualization and accurate presentation is often important to pay more attention.
Abstract (line 21): adding “of these BGCs” does not make it understandable. Fifty-three BGCs are more than half of 261 BGCs?
Line 70: What is the biochemical analysis used in this manuscript?
Table 1. Fonts are not uniform. Hopefully, the editing process will help later on.
Author Response
Here is our point-to-point answer is underlined in blue. Thanks for your help in improving the manuscript. Best regards Comments and Suggestions for Authors
Thank you for the making the effort to revise the manuscript. The main figures have been provided and included now. The figures help me to better understand the story. However, those figures need some work to improve resolutions and readability (e.g. some figure texts are not readily readable). Related to the figures, there are additional comments or questions:
The figures which are unsufficiently resoluted or readable are not specified.
We can not see which figure is not readible. However if the editing process requires it, the font size can be increased in some captions.- Figure 1. (i) How comparable are phylogenetic trees generated based on 16S rRNA gene sequence versus multilocus housekeeping genes as applied in the reference 33? (ii) Is there any reason not to include an outgroup?
- Figure 4. Description on how to determine GCFs as unique ones would be beneficial to have in the manuscript.
Specific comments:
In general, data visualization and accurate presentation is often important to pay more attention.
We cannot consider this remark since there is no specific request and rather question the whole manuscript. Furthermore we improved the manucript rigorously following the requests and provided all the additional data required. This is a general comment appearing in review 2 that was not specified in the first review.
Abstract (line 21): adding “of these BGCs” does not make it understandable. Fifty-three BGCs are more than half of 261 BGCs?
Right. This is clealy a mistake.
The sentence was corrected to become 'A significant part of these BGCs (n=53) were unique ...'
Line 70: What is the biochemical analysis used in this manuscript?
'Biochemical' was replaced by 'metabolomic'.
Table 1. Fonts are not uniform. Hopefully, the editing process will help later on.
Fonts are not intentionally uniform to help distinguishing chromosome from plasmid sequences. If requested they can be homogeneized in size.